# Analysis and Parameter Design of SiC-Based Current Source Inverter (CSI)

**Xingjian Yang** [1], **Zhennan Zhao** [2], **Cheng Wang** [2], **Jianzhi Xu** [3], **Kefu Liu** [1,3,*] and **Jian Qiu** [3,*]

1 Academy for Engineer & Technology, Fudan University, 220, Handan Road, Shanghai 200433, China
2 School of Automation, Nanjing University of Science and Technology, No. 200, Xiaolingwei Street Xuanwu District, Nanjing 210094, China
3 School of Information Science and Engineering, Fudan University, 220, Handan Road, Shanghai 200433, China
* Correspondence: kfliu@fudan.edu.cn (K.L.); jqiu@fudan.edu.cn (J.Q.);
  Tel.: +86-13651784684 (K.L.); +86-15921640969 (J.Q.)

**Abstract:** Current source inverters (CSIs) use inductors as the major component to store energy. Compared with voltage source inverters (VSIs), CSIs have two advantages: 1. They can avoid the converter failure caused by capacitor failures, and 2. The load current does not increase with load mutation or even short-circuit failure. Therefore, CSIs can be a promising technology for EV charging. However, the waveforms, parameter design procedure, and power efficiency are still unclear. Therefore, it is unclear if CSIs are suitable for EV chargers. This article derives the closed-loop equations of the critical components, including the inductor current waveforms and the voltage ripple. Especially, the load over-voltage phenomenon is derived and verified to further ensure the reliability of the CSI system. Based on the derived equations and reliability requirements, the parameter design procedure is proposed. The power efficiency of both the Si- and SiC-based converters are derived and compared to remove the barrier of applying CSIs in EV chargers in the industry. Our simulations and experiments verify the correctness of the system modeling, over-voltage phenomenon, and power efficiency. All the simulation files (using PLECS) and calculation files (using MATLAB) are attached for the readers to verify and/or further modify.

**Keywords:** current source inverter (CSI); inductor-based converter; SiC converter; power efficiency

## 1. Introduction

Source inverters are widely used in advanced electric vehicle (EV) chargers [1–6], grid-tied photovoltaic (PV) systems [6–11], wind turbine generator (WTG) systems [6], motor drives [12,13], data center power supplies [14–17], etc. In EV charging technology, bidirectional vehicle-to-grid and grid-to-vehicle charging can make full use of the internal power cells of EVs and employ them in the microgrid as mobile energy storage units [18]. Figure 1 gives two examples: the grid-connected PV-EV battery charging system [19] and the grid-connected wind turbine generator (WTG)-EV battery charging system [20]. As shown in Figure 1, a bidirectional inverter is necessary for transferring power from PV array to grid because the output of PV generators is DC; and as shown in Figure 1. (b), a bidirectional inverter is used for transferring power from the DC chargers to wind generators because the output of wind generators is AC. Moreover, the bidirectional chargers adopt the reflex charging control strategy [21,22], considering traditional DC–DC charging processes have the problems of incomplete reaction, excessive thermal generation, and a short battery life cycle due to long-lasting charging [23]. Moreover, bipolar pulses produced by inverters can be used in the reflex charging mode [24]. Compared with the traditional DC power charging process, the reflex charging mode can speed the charging process up and extend the life of batteries [21,25–27]. Generally, inverters are classified into voltage source inverters (VSIs) and current source inverters (CSIs) by considering the structure of the inverters [28–31]. Figure 2 shows the typical topologies of VSIs and

CSIs [28]. For VSIs, the DC voltage is an AC/DC rectifier, and a large capacitor is used to maintain the DC-link voltage stability; as for CSIs, the DC current source is an AC/DC rectifier with a large inductor that is used to maintain a constant current [28,29]. VSIs use DC-link capacitors as the major components to store energy, while CSIs use inductors to store energy [31].

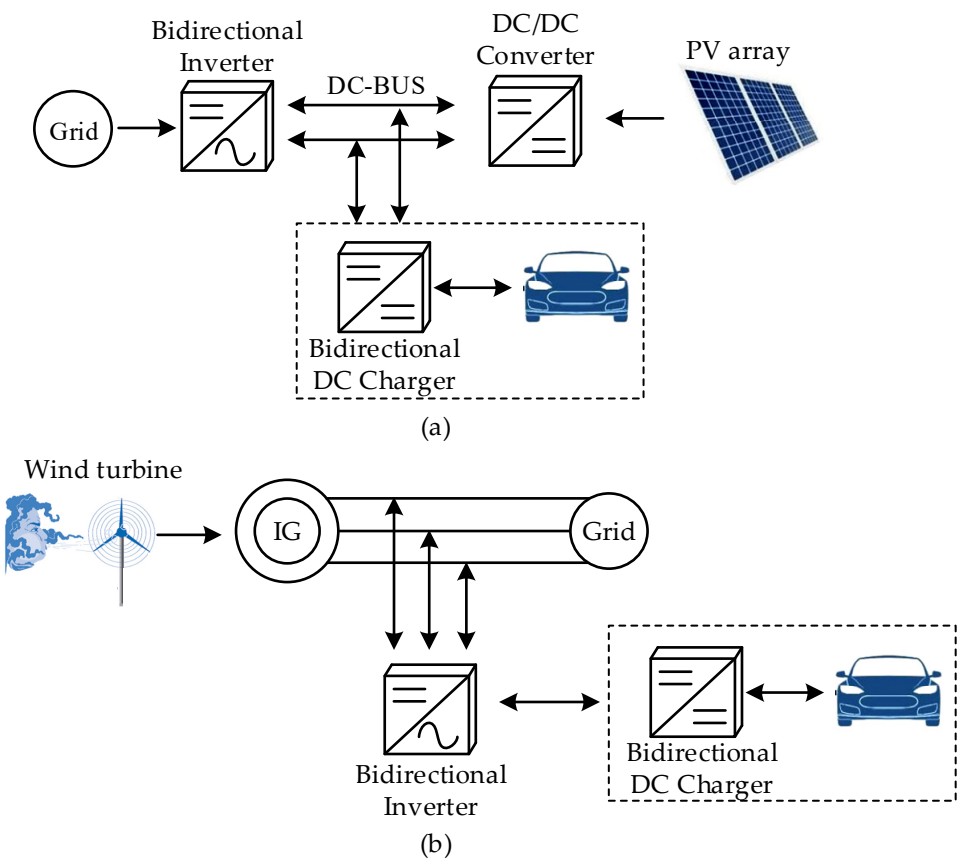

**Figure 1.** (**a**) EV charging system with PV array and grid, and (**b**) WTG-EV battery charging system. Adapted from [19,20].

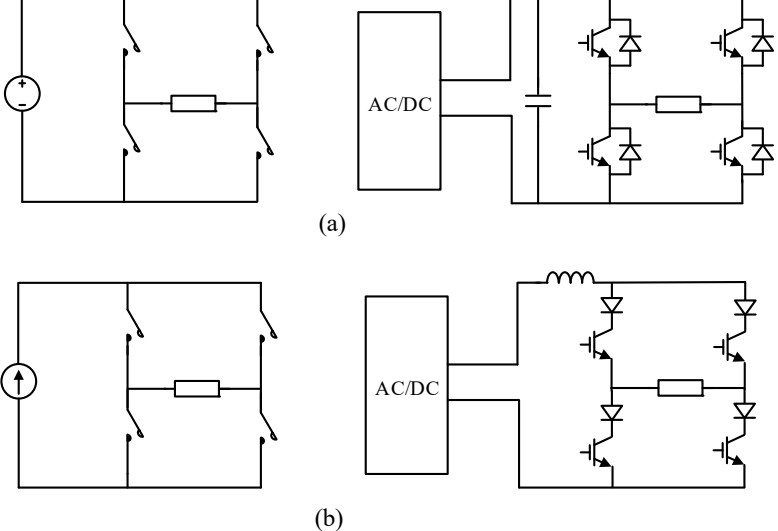

**Figure 2.** Typical topologies of inverters: (**a**) voltage source inverter, and (**b**) current source inverter.

Meanwhile, CSIs are mostly used in reliability-sensitive applications, such as electric field welding [32,33]. The authors of [34] evaluated the mean time between failure (MTBF) and concluded that the MTBF of capacitors could dominate the MTBF of the whole converter. Therefore, medical grade products (e.g., electric field welding [32,33] and electrosurgical generators [35]) still prefer CSIs.

Furthermore, CSIs are also widely used in applications that suffer load mutation [35]. Because CSIs use inductors as the major components to store energy, they tend to keep the current constant during load mutation or even a short-circuit fault without extra high-speed feedback control. Alternatively, because VSIs tend to keep the voltage constant, VSIs suffer from a large current if circuits encounter a short-circuit fault.

These advantages in reliability and short-circuit protection are promising in EV chargers. Firstly, EV chargers are automation grade application and require high reliability. CSIs as capacitorless converters can have a high MTBF [34] and can operate with high environmental temperatures. Furthermore, when the batteries encounter a short-circuit fault, a high current should be strictly forbidden. Therefore, CSIs are inherently suitable for EV chargers and battery protection.

On the other hand, wide band-gap (WBG) semiconductor technologies such as silicon carbide (SiC) and gallium nitride (GaN) recently have received increased attention for their superior characteristics [36–41], as Figure 3 [42] shows. Specifically, applications such as electric vehicles (EVs) demand that WBG semiconductors meet their stringent specifications [42].

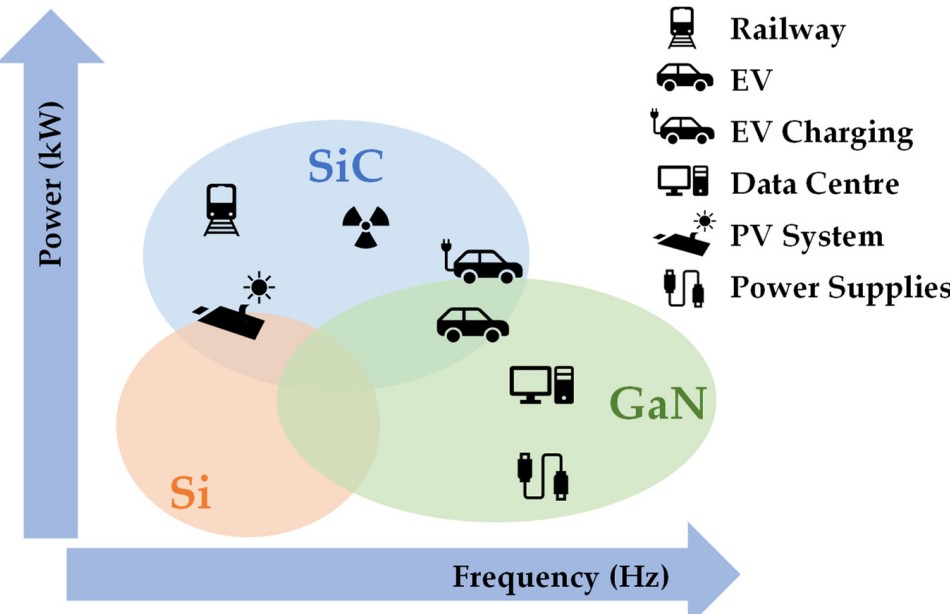

**Figure 3.** Emerging of WBG semiconductor devices in transportation applications. Adapted from [42].

The physical properties of SiC, Si, and GaN materials are given in Table 1 [43], SiC MOSFETs have a higher breakdown electric field, which allows higher and thinner doped voltage blocking layers [44]. It makes SiC MOSFETs have a lower voltage drop and smaller on-state resistance than those of Si at the same voltage rating [45]. In addition, a higher breakdown electric field results in a smaller die area, which makes the junction capacitance of SiC MOSFETs smaller than that of Si MOSFETs. Therefore, SiC MOSFETs have a reduced switching loss. Therefore, replacing Si MOSFETs with SiC MOSFETs can improve the efficiency of the inverter.

**Table 1.** Physical Properties of SiC, GaN, and Si materials.

| Electrical Property | Si | SiC | GaN |
|---|---|---|---|
| Band Gap Energy (eV) | 1.1 | 3.26 | 3.4 |
| Electric Field ($\times 10^6$ V/cm) | 0.3 | 3 | 3.5 |
| Electron Mobility ($\times 10^3$ cm$^2$/V·s) | 1.3 | 0.9 | 0.9–2 |
| Thermal Conductivity (W/cm·K) | 1.5 | 3.7 | 1.3 |
| Saturation Drift Velocity ($\times 10^7$ cm/s) | 1.0 | 2.0 | 2.5 |

Figure 4 shows the structure of a cell of Si MOSFET, SiC MOSFET and GaN HEMT [46,47]. The SiC MOSFET is a vertical trench construction similar to Si MOSFETs, while GaN is a lateral construction [46,47]. Therefore, SiC parts are usually available in a compatible package style, such as TO-247 and TO-220, allowing them to drop in as replacements for MOSFETs and IGBTs in existing designs, giving immediate advantages [47]. While GaN HEMTs have a different structure and mechanism, they still need further analysis on aspects such as dynamic on-state resistance [48,49]. Therefore, applications with high reliability requirements prefer SiC MOSFETs.

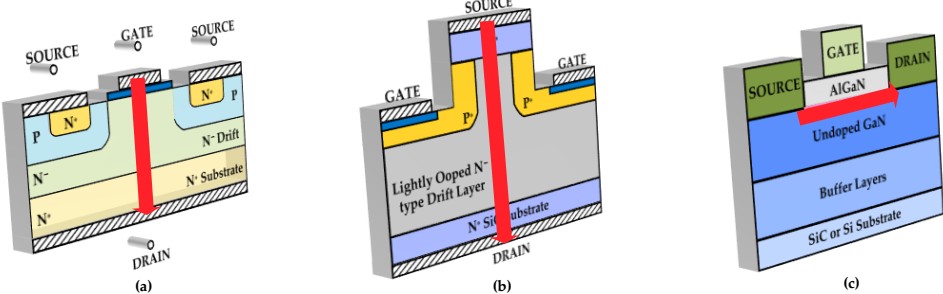

**Figure 4.** The structures of (**a**) Si MOSFET, (**b**) SiC MOSFET, and (**c**) GaN HEMT. [46] Adapted from [46,47].

The authors of [35] proved CSIs' ability in the transition between the three control modes: the constant voltage, current, and power modes. Therefore, [35] forms the foundation for the CSIs' application in EV chargers. The authors of [24] demonstrated that CSIs can be applied to EV chargers and proposed a drive circuit to prevent output voltage overshoot. However, two basic questions remain that prohibit CSIs' application in the industry: 1. The parameter design procedure is unclear; 2. The power loss is unclear. Especially with SiC MOSFETs' wide application, both the volume and power loss of CSIs would be reduced. Theoretic analyses would be important to discuss the possibility to use CSI topology as the EV charger.

This paper comprehensively derives the expressions of the critical waveforms in CSIs. Wherein, the output voltage and its ripple are critical for overvoltage protection; the inductor current and its ripple can be used to calculate the volume and power loss of the inductors. Therefore, the comprehensive parameter design diagram is given. Furthermore, the power loss as well as the power efficiency calculation is presented. With the power loss equations, this paper also compares the power efficiency between the converters with Si MOSFETs and the converters with SiC MOSFETs, and hence verifies the advantages of applying SiC MOSFETs in CSIs.

In this research, the circuit operating condition analysis and the formulas of output characteristics were studied, including the output voltage and its ripple, which are necessary for proper overvoltage protection design. The parameter design is presented as well. Moreover, the power loss calculation and efficiency estimation are discussed, especially for its application based on SiC power devices, providing the possibility for its application in the industry. Finally, the correctness of those theoretical analyses was verified through both simulations and experiments.

The rest of the article is organized as follows: Section 2 presents the circuit topology and the operation principles. With the derived expressions, the article analyzes the reason for the voltage overshoot and the magnitude and shape of the current ripple. Furthermore, Section 2 discusses the parameter design procedure. Section 3 derives the equation of the power conversion efficiency and compares the power efficiency between Si- and SiC-based CSIs. Furthermore, Section 4 presents the simulations and experiments that verified the waveforms and the power conversion efficiency. Finally, Section 5 concludes the article. And all parameters in this paper are symbolized according to the following nomenclature, as shown in Table 2.

**Table 2.** Nomenclature.

| Symbol | Parameter |
|---|---|
| $L$ | Power Inductor |
| $S_1 \sim S_5$ | Switches |
| $U_{DC}$ | Input DC Voltage |
| $R_{load}$ | Load Impedance |
| $I_L$ | Average Inductor Current |
| $I_{L.\max}$ | Inductor Current Peak Value |
| $I_{L.\min}$ | Inductor Current Valley Value |
| $\Delta i_L$ | Inductor Current Ripple |
| $f_s$ | Switching Frequency of $S_5$ |
| $T_s$ | Inverter Working Period |
| $f_{sw}$ | Switching Frequency of $S_1 - S_4$ |
| $D$ | Duty Cycle of $S_5$ |
| $R_{ds(on)}$ | MOSFET On-State Resistance |
| $V_F$ | Diode Forward Voltage |
| $p_{con}$ | MOSFET Conduction Loss |
| $p_{swi}$ | MOSFET Switching Loss |
| $E_{on}$ | MOSFET Turn-On Switching Energy |
| $E_{off}$ | MOSFET Turn-Off Switching Energy |
| $E_{swi}$ | MOSFET Switching Energy |

## 2. Operation and Modeling of CSIs

The topology of the capacitor-free CSI is shown in Figure 5 [24]. Ref. [24] describes the operation principle of the circuit but could not provide the detailed waveform equations. Section 2 presents the circuit, operation principles, and modulation scheme, and then derives the waveform equations with both the simplified and precise models. Moreover, the difference between the precise and accurate models is compared. Based on all the equations, the parameter design procedure is proposed as well.

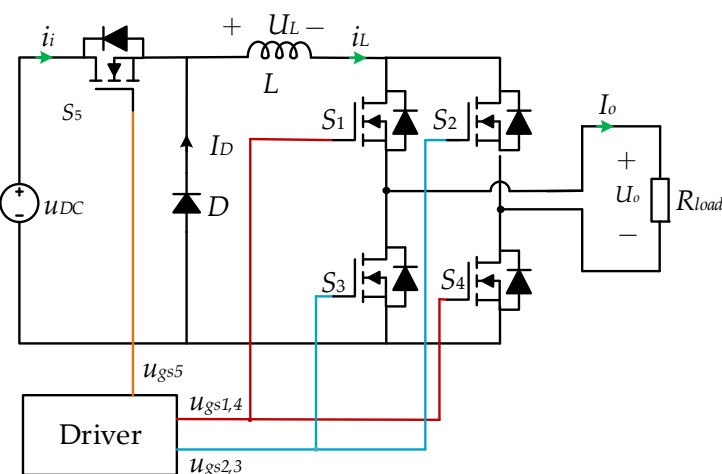

**Figure 5.** Topology of the capacitor-free inverter.

### 2.1. Circuits, Operation Principles, and Modulation Scheme

As shown in Figure 6, the CSI was operated in switching period $T_s$ corresponding to the frequency $f_s$ of $S_5$, and $S_1$–$S_4$ operated at a higher frequency $f_{sw}$ to produce bipolar pulses. The $D$ denotes the duty cycle of $S_5$. Under this operation scheme in Figure 6, four circuit stages of the inverter are shown in Figure 7. It also shows the current flow in each mode.

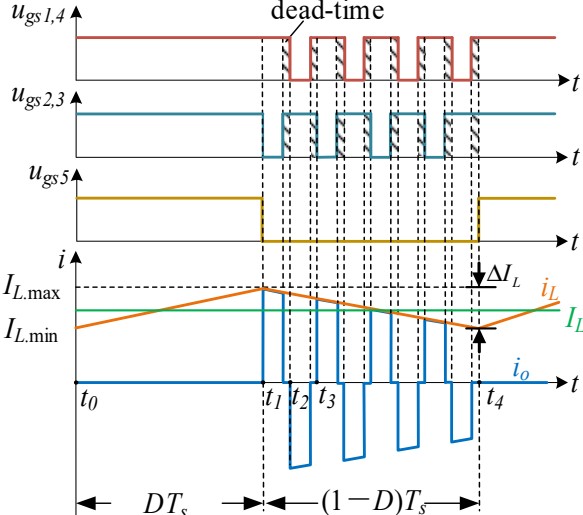

**Figure 6.** Sequence diagram of the proposed inverter.

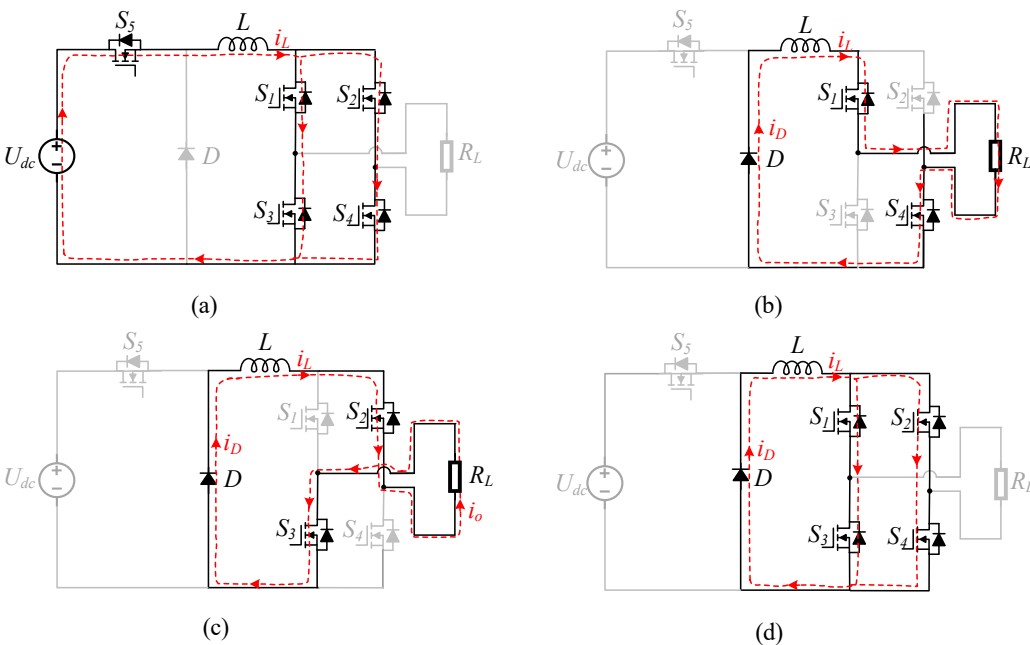

**Figure 7.** Inverter topology transformation among different stages: (**a**) the charging stage; (**b**) positive pulse; (**c**) negative pulse; (**d**) deadtime.

### 2.2. Operation Principles and Two Models

#### 2.2.1. Simplified Model

Firstly, a simplified model was constructed with ideal switches. In this model, the ideal switches were short circuit in the on-state and open circuit in the off-state.

(1)　The charging stage: $(t_0 - t_1)$

As shown in Figure 7a, when $S_1$–$S_5$ were ON and the freewheel diode was OFF, the inductor was charged by the input DC source. Because $S_5$ was modeled as an ideal switch, there was no voltage across it. The circuit voltage equation based on KVL is:

$$v_L(t) = U_{DC} = L\frac{di_L}{dt} = L\frac{\Delta i_L}{DT_s} \tag{1}$$

where $U_{DC}$ is the input DC voltage, and $\Delta i_L$ is the inductor current ripple defined as the difference between the inductor current peak and valley values. The decrease of the inductor current was equal to the increase of the inductor when the inverter worked in a steady state and can be expressed as:

$$\Delta i_L = \frac{DU_{DC}}{Lf_s} \tag{2}$$

(2)　The discharging stage: $(t_1$–$t_4)$

As shown in Figure 7b,c, $S_5$ was always OFF in this stage, and $S_1$, $S_4$ and $S_2$, $S_3$ were two pairs of switches operating in complementary control. Since the freewheel diode and $S_1$–$S_4$ were modeled as ideal switches, the circuit voltage equation based on KVL is:

$$v_L(t) + v_o(t) = 0 \tag{3}$$

The average voltage across inductors was zero when the circuit was in a steady state:

$$\frac{1}{T_s}\int_0^{T_s} v_L(t)dt = \frac{1}{T_s}\int_0^{DT_s} U_{dc}dt + \frac{1}{T_s}\int_0^{(1-D)T_s} -v_{load}(t)dt = 0 \tag{4}$$

$$U_{DC} \cdot DT_s - I_L R_{load}(1-D)T_s = 0 \tag{5}$$

Moreover, the average current of inductor $I_L$ can be expressed as:

$$I_L = \frac{DU_{DC}}{(1-D)R_{load}} \tag{6}$$

### 2.2.2. Precise Model

This section proposes a precise model where the MOSFETs were modeled as a constant resistor $R_{ds(on)}$, and on-state and freewheel diodes were modeled as a constant voltage drop $V_F$ in the on-state. They were both modeled as an open circuit in the off-state. The equivalent circuit of the charging and discharging states are shown in Figure 8.

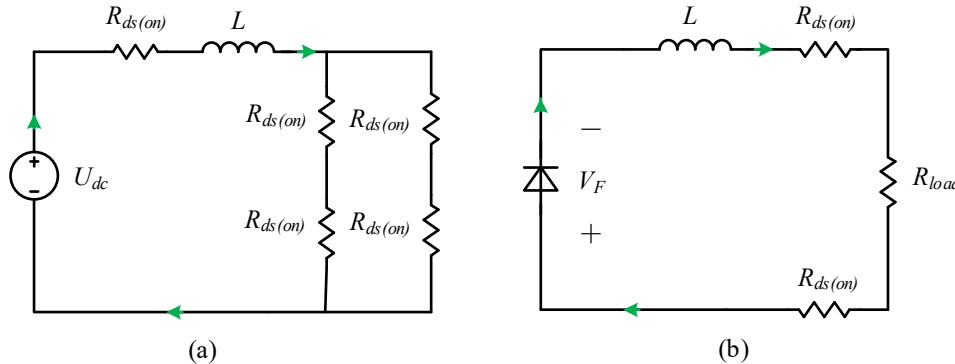

(a)　　　　　　　　　　　　　　　　(b)

**Figure 8.** The equivalent circuit: (**a**) the charging state, and (**b**) the discharging states.

As shown in Figure 8, both stages can be described as an *R–L* fist-order circuit with one excitation, and the differential equation that describes the inductor current can be expressed as (7) and (8):

$$L\frac{di_L}{dt} + 2R_{ds(on)}i_L = U_{DC} \tag{7}$$

$$L\frac{di_L}{dt} + i_L(2R_{dson} + R_{load}) = -V_F \tag{8}$$

The peak and valley values of the inductor current are denoted as $I_{l.\max}$ and $I_{L.\min}$, respectively. By solving the differential Equations (7) and (8), the inductor currents in the charging and discharging states can be expressed as (9) and (10), respectively:

$$i_{L\_Ch}(t) = I_{L.\min}e^{-2R_{dson}t/L} + \frac{U_{DC}}{2R_{dson}}\left(1 - e^{-2R_{dson}t/L}\right) \tag{9}$$

$$i_{L\_Dis}(t) = I_{L.\max}e^{-(2R_{dson}+R_{load})t/L} + \frac{-V_F}{2R_{dson}+R_{load}}\left(1 - e^{-(2R_{dson}+R_{load})t/L}\right) \tag{10}$$

The energy discharged by the inductor and charged from the inductor reached a balance when the inverter worked in the steady state, as shown in (11) and (12), respectively:

$$|\Delta i_{L+}| = |\Delta i_{L-}| \tag{11}$$

$$i_{L\_Ch}(DT_c) - i_{L\_Ch}(0) = i_{L\_Dis}(0) - i_{L\_Dis}((1-D)T_c) \tag{12}$$

By solving (11), the peak and valley values of the inductor currents $I_{L.\max}$ and $I_{L.\min}$ can be obtained according to Equations (13) and (14), respectively:

$$I_{L.\min} = \frac{\frac{U_{dc}e^{-(2R_{ds(on)}+R_{load})(1-D)/(Lf_s)}}{2R_{ds(on)}}\left(1 - e^{-2R_{ds(on)}D/(Lf_s)}\right) - \frac{V_Fe^{-(2R_{ds(on)}+R_{load})(1-D)/(Lf_s)}}{2R_{ds(on)}+R_{load}}}{1 - e^{-2R_{ds(on)}(D/(Lf_s)-(R_{load}+2R_{ds(on)})(1-D)/(Lf_s))}} \tag{13}$$

$$I_{L.\max} = I_{L.\min}e^{-2R_{ds(on)}D/(Lf_s)} + \frac{U_{dc}}{2R_{dson}}\left(1 - e^{-2R_{ds(on)}D/(Lf_s)}\right) \tag{14}$$

### 2.3. Model Comparison and Error Analysis

The simplified model was based on the condition that all switches were ideal switches, while in the precise model, the diode was modeled as constant voltage $V_F$ and the MOSFET was modeled as resistor $R_{ds(on)}$. Therefore, the simplified model was valid for all kinds of switches (SiC MOSFETs, Si MOSFETs, and GaN HEMTs). However, GaN HEMTs' dynamic on-state resistance was different compared with those of MOSFETs. Therefore, the precise model was valid for SiC and Si MOSFETs, but less valid for GaN HEMTs. The errors of the two models are discussed below.

Figure 9a,b show the comparison of the inductor current waveforms between simulations, the simplified model, and the precise model. As shown in Figure 9a, the results calculated by the simplified and precise models both matched the simulation results when the inverter was on the rated conditions. However, when the inverter was under a low load, as shown in Figure 9b, the error of the results calculated by the simplified model was great, but the results calculated by this precise model matched the simulation results very well.

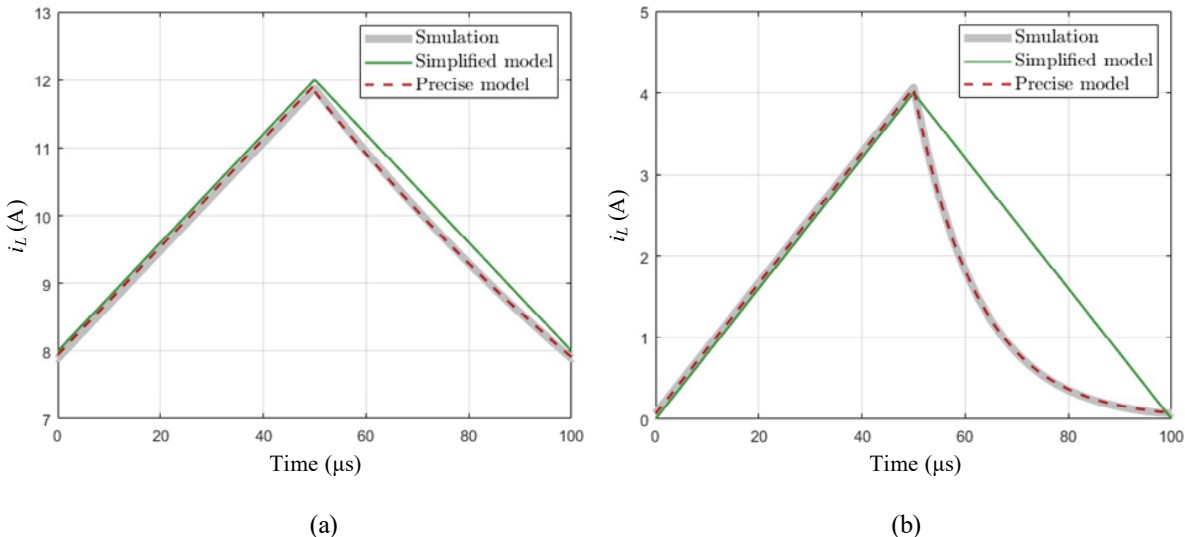

**Figure 9.** The comparison of the inductor current waveforms between simulations, the simplified model, and the precise model: (**a**) when $R_{load}$ = 10 Ω, and (**b**) $R_{load}$ = 100 Ω.

The error of the inductor current ripple $\Delta i_L$ and the inductor average current $I_L$ solved by the proposed two models are shown in Figure 10, where $I_L$* and $\Delta i_L$ * were solved by the simplified model, and $I_L$ and $\Delta i_L$ were solved by the precise model. As shown in Figure 10, the error between these two models increased with the increase of $I_L$, and the error was positively correlated with $R_{DS(on)}$. The error of $I_L$ and $\Delta i_L$ were both within 10% when $R_{ds(on)}$ was less than 500 mΩ and $I_L$ was less than 30 A. When the $R_{ds(on)}$ of the MOSFETs was smaller than one milliohm of resistance, the error of $I_L$ and $\Delta i_L$ were both less than 5%.

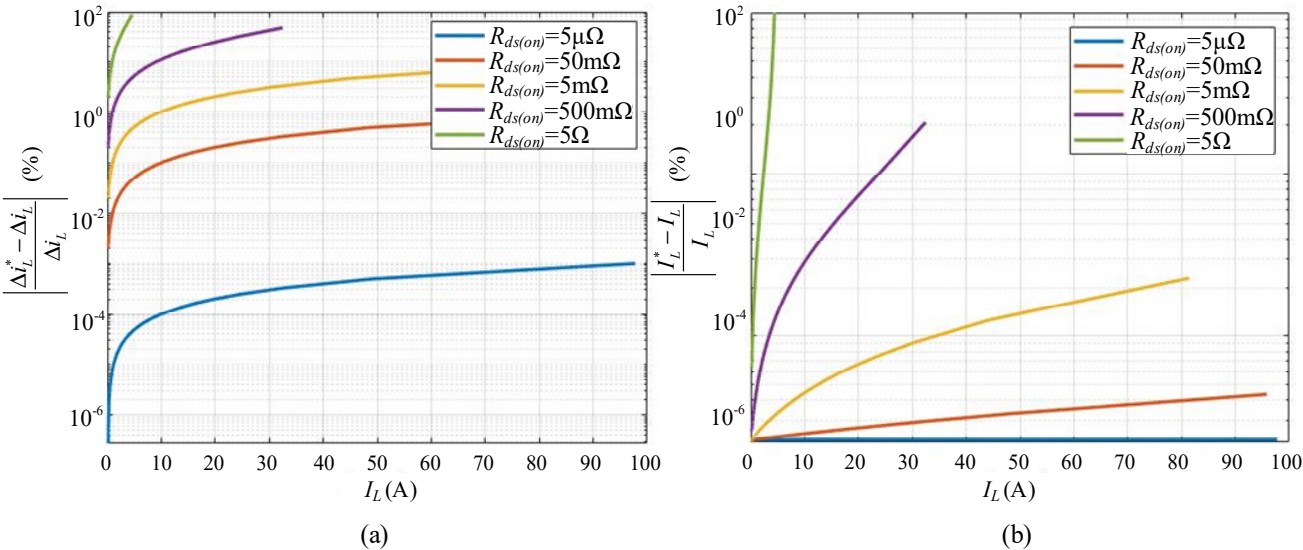

**Figure 10.** The error between the simplified and precise model: (**a**) the error of $\Delta i_L$, and (**b**) the error of $I_L$.

In the precise model, MOSFETs were modeled as a constant resistor, but the MOSFETs' resistance changed with $I_D$. However, this variation was small, especially for SiC MOSFETs (within 16.67%). With the parameter variation shown in Table 3 [50,51], simulation results with a typical, minimum, and maximum $R_{ds(on)}$ were conducted. The results are shown in Figure 11. It indicated that the precise model was still valid when $R_{ds(on)}$ changed with $I_D$.

**Table 3.** The comparison between SiC and Si MOSFETs.

| $R_{ds(on)}$ @25 °C | SiC MOSFET (C3M0025065D, $V_{gs}$ = 15 V) | Si MOSFET (IPW60R037P7, $V_{gs}$ = 10V) |
|---|---|---|
| Minimum | 24 ($I_D$ = 10A) | 60 ($I_D$ = 0 A) |
| Typical Value | 25 ($I_D$ = 33.5A) | 37 ($I_D$ = 29.5 A) |
| Maximum | 28 ($I_D$ = 135A) | 95 ($I_D$ = 135 A) |

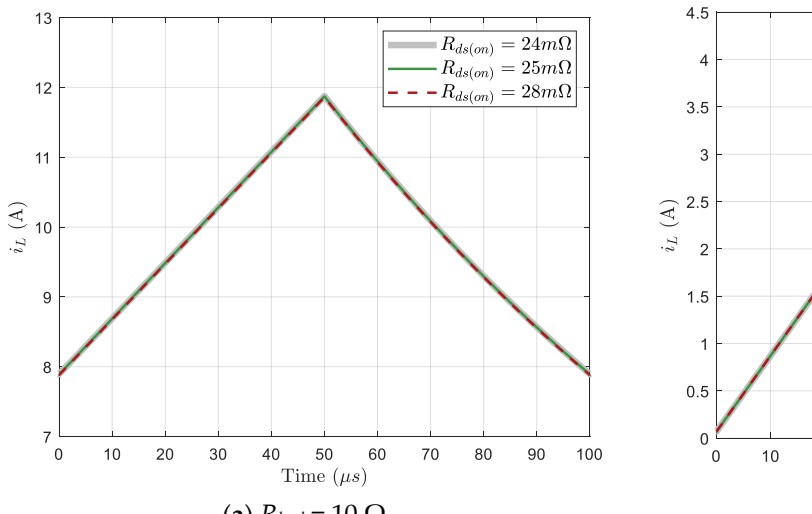

**(a)** $R_{load}$ = 10 Ω

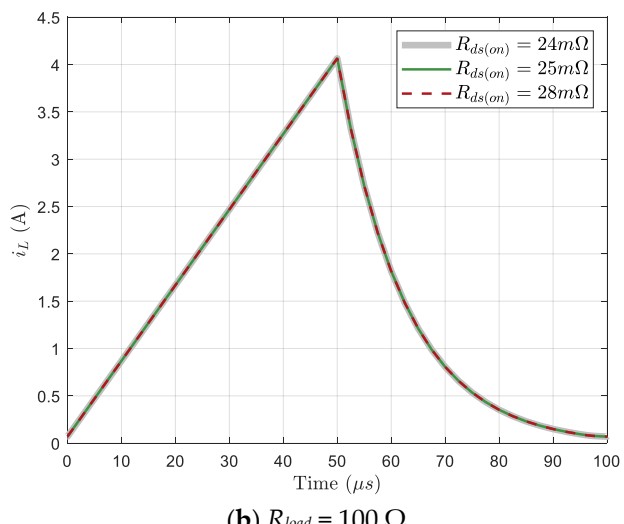

**(b)** $R_{load}$ = 100 Ω

**Figure 11.** Simulation results: (**a**) when $R_{load}$ = 10 Ω, and (**b**) $R_{load}$ = 100 Ω.

*2.4. Parameter Design Based on Simplified Model*

2.4.1. Inductor Design

Inverter Working in CCM Mode

Because CSIs work in the CCM mode, $i_L$ is always larger than zero, and its minimal value $I_{L.min}$ is zero:

$$I_{L.\min} = I_L - \frac{1}{2}\Delta I_L = 0 \tag{15}$$

$$L_c = \frac{(1-D)R_{load}}{2f_s} \tag{16}$$

The Current Ripple

The current ripple was calculated by (2). As shown in (2), the inductor current ripple $\Delta i_L$ was determined by the inductance when the input DC voltage $U_{DC}$ was constant and the charge–discharge frequency $f_s$ and duty cycle $D$ were fixed. The current ripple was reversely correlated with the inductance. To ensure that the ripple of the inductance current met the ripple current limits within the full input voltage range, the maximum value of inductance under different modes was:

$$\Delta i_L = \frac{DU_{DC}}{Lf_s} \leq \gamma I_{L.\max} \tag{17}$$

where $\gamma$ is the ripple quotient defined as the ratio of $\Delta i_L$ to $I_L$.

The inductance should be designed to satisfy both the above conditions:

$$L \geq \max\{L_c, L_{\min}\} \tag{18}$$

#### 2.4.2. Active Switches Selection

Voltage Stress

During the discharge state, $S_5$ was turned off, as shown in Figure 7b–d, and the voltage potential of its drain is $U_{dc}$; the voltage potential of its source is $-V_F$, so the voltage stress of $S_5$ was:

$$V_{DS,S_5} = U_{dc} + V_F \tag{19}$$

During the discharge state, the voltage of the turn-off switch was the sum of the voltage of load resistance and the conduction voltage of the on-state switch, so the voltage stress of $S_1$–$S_4$ can be expressed as:

$$v_{DS,S_1 \sim S_4} = i_L \cdot \left( R_{Load} + R_{DS(on)} \right) \tag{20}$$

From the previous analysis, the inductance current decreased during the discharging state; thus, the maximum $V_{DS}$ was at the start point of the discharge state:

$$V_{DS.\max(S_1 \sim S_4)} = I_{L.\max} \cdot \left( R_{load} + R_{DS(on)} \right) \geq U_{load.\max} \tag{21}$$

Current Capability of Switches

The drain source current of $S_5$ was equal to the inductance current during the charge state, and could be turned off during the discharge state, so the average current in each charging period was:

$$I_{D(pulse),S_5} = I_{L.\max} \tag{22}$$

$$I_{D(av),S_5} = \frac{1}{T} \int_0^{DT_s} (i_L) dt = D I_L \tag{23}$$

The active switches $S_1$–$S_4$ were always ON during the charge state, and two pairs of switches were alternatively turned on and off during the discharge state with a high frequency $f_{sw}$. $S_1$ and $S_4$ were turned on to output a positive pulse; $S_2$ and $S_3$ were turned on to output a negative pulse if the output pulse widths of the positive and negative pulses were equal, i.e., the conduction times of $S_1/S_4$ and $S_2/S_3$ were equal to half the discharge state:

$$I_{D(S_1 \sim S_4)} = \frac{1}{T_s} \left[ \int_0^{DT_s} \left( \frac{1}{2} i_L \right) dt + \int_0^{(1-D)T_s} (i_L) dt \right] = \frac{1+D}{2} I_L \tag{24}$$

#### 2.4.3. Freewheeling Diode Selection

During the charging state, as Figure 7a shows, the diode was turned off by the reverse voltage, $v_{reverse}$:

$$v_{reverse} = U_{DC} - i_L \cdot R_{DS(on)} \tag{25}$$

The inductance current increased during the charge state, and thus $V_R$ reached its maximum at the end point of the discharge state and fell to its minimum when the inductance charging finish time was:

$$V_{R.\max} = U_{DC} - I_{L.\min} \cdot R_{DS(on)} \tag{26}$$

The diode could only be turned on during the discharge state and when its current was equal to the inductance current.

The average current in each charging period was:

$$I_F = \frac{1}{T_s} \int_0^{(1-D)T_s} (i_L) dt = (1-D) I_L \tag{27}$$

### 3. Power Loss Estimation Based on the Precise Model

*3.1. Conduction Loss*

The conduction losses of MOSFETs were usually expressed using their on-state resistance $R_{ds(on)}$, while the conduction losses of freewheel diodes were usually expressed using their forward threshold voltage $V_F$. Moreover, from the analysis in Section 2.2, we know there was current flowing through $S_5$ only in the charging state, and this current flowed through two circuit branches consisting of $S_1/S_3$ and $S_2/S_4$ on average. However, there was current flowing through the freewheel diode only in the discharging state, and this current also flowed through $S_1/S_4$ or $S_2/S_3$ by turns. Therefore, the conduction of each switch can be expressed as:

$$p_{cond,S_5} = \left( \int_0^{DT_s} i_{L\_Ch}^2 R_{ds(on)} dt \right) f_s \tag{28}$$

$$p_{cond,S_1=S_2=S_3=S_4} = \left[ \int_0^{DT_s} \left( \frac{i_{L\_Ch}}{2} \right)^2 R_{DS(on)} dt + \frac{1}{2} \int_0^{(1-D)T_s} i_{L\_Dis}^2 R_{ds(on)} dt \right] f_s \tag{29}$$

$$p_{cond,Diode} = \left( \int_0^{(1-D)T_s} V_F i_{L\_Dis} dt \right) f_s \tag{30}$$

*3.2. Switching Loss*

The switching process of MOSFETs took some time when the voltage and current waveforms overlapped, which produced the switching loss. Therefore, the energy loss can be expressed as:

$$p_{sw} = \left( \langle E_{on} \rangle_{T_s} + \langle E_{off} \rangle_{T_s} \right) f_s = \left[ \int_0^{t_{on}} u_1(t) i_1(t) dt + \int_0^{t_{off}} u_2(t) i_2(t) dt \right] f_s \tag{31}$$

where $t_{on}$ is the time interval of the turn-on stage, while $t_{off}$ is the time interval of the turn-off stage. Moreover, $u_1$ and $i_1$ are the voltage and current of MOSFETs when they are turned on, respectively, while $u_2$ and $i_2$ are the voltage and current of MOSFETs when they are turned off, respectively.

Here, parasitic inductances of MOSFETs were not taken into consideration, and the switching behavior was determined by the parasitic capacitance consisting of $C_{gd}$, $C_{gs}$, and $C_{ds}$.

Figure 12 shows the waveforms when the MOSFETs were turning on and off. Combining (30), the power loss of the turn-on and -off stages, respectively, can be calculated as:

$$\begin{aligned} E_{on} &= \int_0^{t_{ri}} V_{DS1} \left( \frac{I_1}{t_{ri}} t \right) dt + \int_0^{t_{rr}} V_{DS1} \left( \frac{I_{rr}}{t_{rr}} t \right) dt + \int_0^{t_{fu}} \left( V_{DS1} - \frac{V_{DS1}}{t_{fu}} t \right) I_1 dt \\ &= \frac{1}{2} V_{DS1} I_1 \left( t_{ri} + t_{fu} \right) + \frac{1}{2} V_{DS1} I_{RR} t_{rr} \end{aligned} \tag{32}$$

$$E_{off} = \int_0^{t_{ru}} \left( \frac{V_{DS2}}{t_{ru}} t \right) I_2 dt + \int_0^{t_{fi}} V_{DS2} \left( I_2 - \frac{I_2}{t_{fi}} t \right) dt = \frac{1}{2} V_{DS2} I_2 \left( t_{ru} + t_{fi} \right) \tag{33}$$

where $t_{ri}$ and $t_{fu}$ are the times of current rising and voltage falling during the turn-on transient, respectively, and $t_{ru}$ and $t_{fi}$ are the times of voltage rising and current falling during the turn-off transient, respectively, which can be calculated from (32)–(35).

$$t_{ri} = R_g C_{iss} \ln \left( \frac{V_g}{V_g - V_m} \right) \tag{34}$$

$$t_{fu} = R_g C_{gd,av} \frac{V_{DS} - V_{on}}{V_g - V_m} \tag{35}$$

$$t_{ru} = R_g C_{gd,av} \frac{V_{DS} - V_{on}}{V_m} \tag{36}$$

$$t_{fi} = R_g C_{iss} \frac{V_m - V_{th}}{V_m} \tag{37}$$

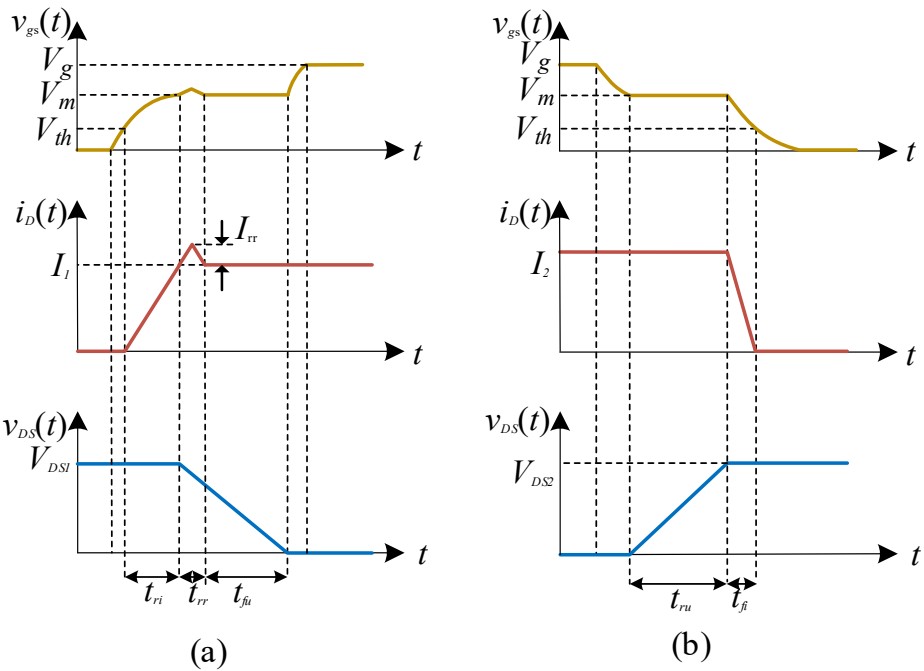

(a)　　　　　　　　　　　　　　　　　　　(b)

**Figure 12.** Waveforms when MOSFETs turned on and off: (**a**) the turning on transient; (**b**) the turning off transient.

(a)　　Analysis of Communication between the Charging and Discharging States

When the CSI transited between the charging and discharging states, $S_1$–$S_4$ stayed ON and there was no power loss consumed from $S_1$–$S_4$ during this process. The current communication was between the freewheel diode and $S_5$, as shown in Figure 13.

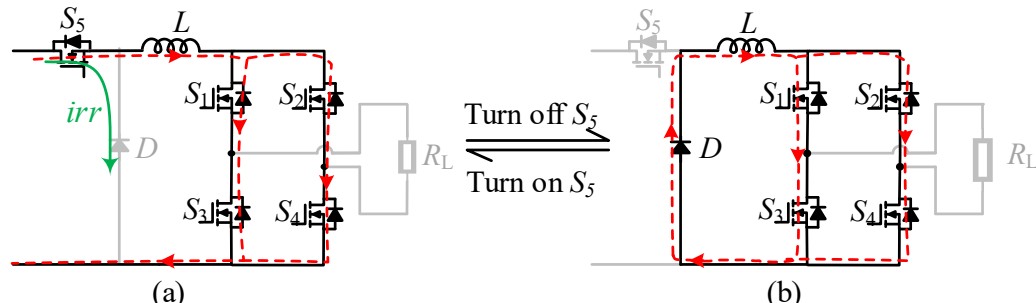

(a)　　　　　　　　　　　　　　　　　　　(b)

**Figure 13.** The communication between the charging and discharging states: (**a**) the charging state, and (**b**) the dead-time of the discharging state.

$S_5$ stayed ON in the charging state when $I_L$ increased, and $S_5$ stayed OFF in the discharging state when $I_L$ decreased. Therefore, as for S$_5$, $I_1$ and $I_2$ in (30) and (31) were the minimum values of the inductor current $I_{L.min}$ and the maximum of inductor current $I_{L.max}$, respectively, and $V_{DS}$ can be calculated with (17). Then, the switching loss of $S_5$ can be calculated as:

$$p_{sw,S_5} = f_s \cdot \left\{ \frac{1}{2}(V_{DC} + V_F)\left[ I_{L.\min}\left(t_{ri} + t_{fu}\right) + I_{L.\max}\left(t_{ru} + t_{fi}\right) \right] + Q_g V_G \right\} \tag{38}$$

where $I_{L.\min}$ and $I_{L.\max}$ can be calculated from (11) in the precise model.

(b)  Analysis of Communication between the P- and N-Modes during the Discharging State

During the discharging state, $S_1/S_4$ and $S_2/S_3$ turned on and off when the working state changed between the P- and N-modes by turns. As shown in Figure 14, the current did not pass through the body diode of $S_1$–$S_4$ in this interval; thus, there was no reverse recovery effect for $S_1$–$S_4$.

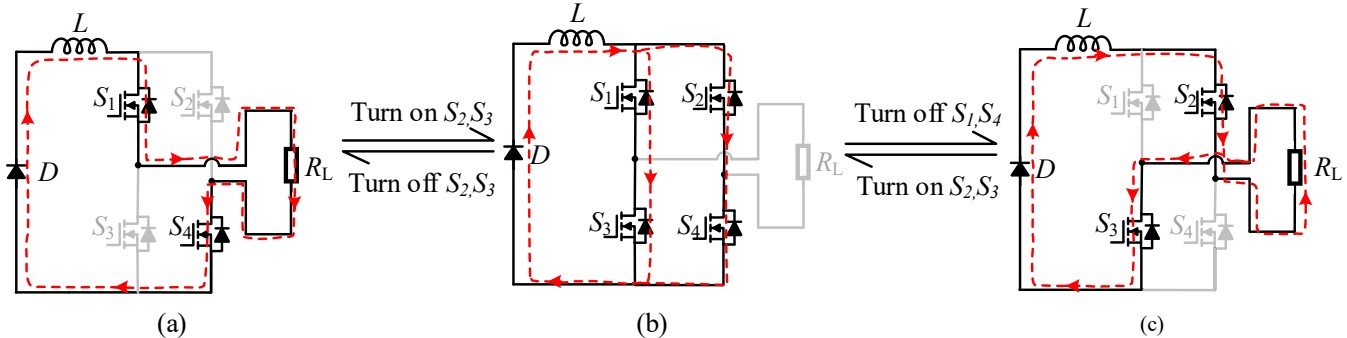

**Figure 14.** The communication between the P- and N-modes during the discharge state: (**a**) the P-mode of the discharge state; (**b**) dead-time of the discharge state, and (**c**) N-mode of the discharge state.

$N_g$ denotes the number of bipolar pulses in one discharging state, so $S_1$ and $S_4$ turned on and off for $N_g$ times. The current of $S_1$–$S_4$ was the current of the inductor that changed during the discharging state given in (6), so the switching loss for each time was different and should be calculated respectively. For the $k_{th}$ switching stage, the switching loss of $S_1$–$S_4$ can be calculated as:

$$E^{(k)}_{sw,S_1=S_4} = \frac{1}{2}\left(R_{load}+R_{ds(on)}\right)\left[i^2_{L(t=k/f_{sw})}\left(t_{ri}+t_{fu}\right)+i^2_{L(t=(0.5+k)/f_{sw})}\left(t_{ru}+t_{fi}\right)\right]+Q_gV_G \tag{39}$$

$$E^{(k)}_{sw,S_2=S_3} = \frac{1}{2}\left(R_{load}+R_{ds(on)}\right)\left[i^2_{L(t=(0.5+k)/f_{sw})}\left(t_{ri}+t_{fu}\right)+i^2_{L(t=(k-1)/f_{sw})}\left(t_{ru}+t_{fi}\right)\right]+Q_gV_G \tag{40}$$

Moreover, the total switching loss of $S_1$–$S_4$ in a period of $T_s$ can be calculated as:

$$p_{sw,S_1=S_4} = f_s\cdot\left\{\left(R_{load}+R_{ds(on)}\right)\sum_{k=1}^{Ng}\left[i^2_{L(t=k/f_{sw})}\left(t_{ri}+t_{fu}\right)+i^2_{L(t=(0.5+k)/f_{sw})}\left(t_{ru}+t_{fi}\right)\right]+Q_gV_G\right\} \tag{41}$$

$$p_{sw,S_2=S_3} = f_s\cdot\left\{\left(R_{load}+R_{ds(on)}\right)\sum_{k=1}^{Ng}\left[i^2_{L(t=(0.5+k)/f_{sw})}\left(t_{ri}+t_{fu}\right)+i^2_{L(t=(k-1)/f_{sw})}\left(t_{ru}+t_{fi}\right)\right]+Q_gV_G\right\} \tag{42}$$

Then, the total loss for the CSI can be calculated as:

$$p_{loss-total} = p_{cond,S_5}+4p_{cond,S_1=S_2=S_3=S_4}+p_{cond,Diode}+p_{sw,S_5}+2p_{sw,S_1=S_4}+2p_{sw,S_2=S_3} \tag{43}$$

### 3.3. Efficiency Comparison between Si- and SiC-Based CSIs

To identify the main cause of losses in the inverter, two kinds of power MOSFETs were used in the proposed topology, and their switching and conduction losses were compared. The main electrical characteristics of these two devices are shown in Table 4.

**Table 4.** Main electrical characteristics of power devices.

| Parameter | SiC MOSFET CREE C3M0025065D | Si MOSFET Infineon IPW60R037P7 |
|---|---|---|
| $V_{ds}$ (V) | 650 | 650 |
| $_d$ (A) | 97 | 76 |
| $I_{d(pulse)}$ (A) | 251 | 280 |
| $R_{dson}$ (mΩ) | 25 | 37 |
| $C_{iss}$ (pF) | 2980 | 5243 |
| $C_{oss}$ (pF) | 178 | 85 |
| $C_{rss}$ (pF) | 12 | 156 |
| $Q_g$ (nC) | 108 | 121 |
| $t_{rr}$ (ns) | 51 | 300 |
| $T_j, T_{stg}$ (°C) | −55~150 | −40~175 |
| Price @ quantity = 1 ($) [Note 1] | 27.87 | 13.92 |

Note 1: the prices were obtained from Digikey.com on 21 September 2022.

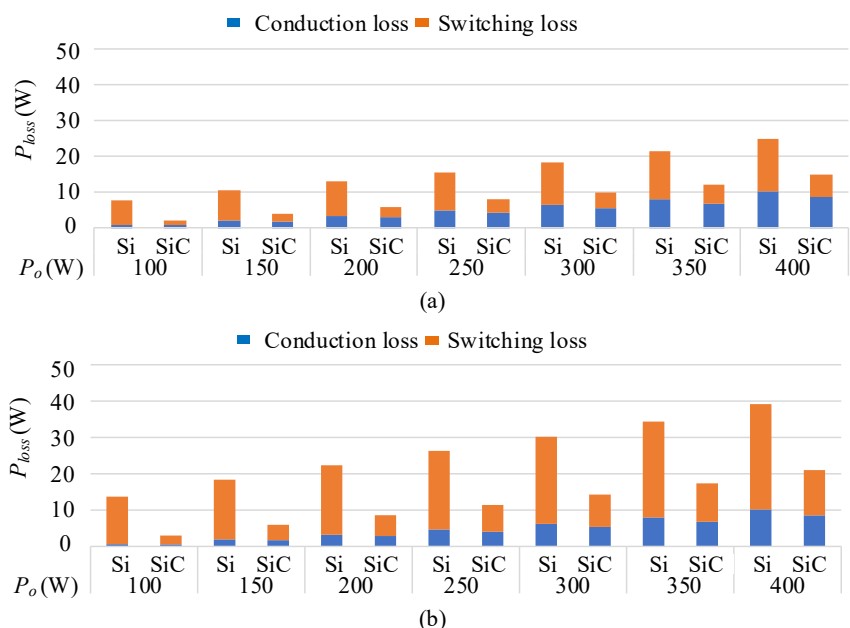

**Figure 15.** The comparative calculation results of the total power loss of CSI based on Si and SiC power devices: (**a**) $f_s$ = 10 kHz, and (**b**) $f_s$ = 20 kHz.

Figure 15 compares the power losses between the SiC- and the Si-based CSIs. The input DC voltage was specified as 100 V, while the output power and the switching frequency varied in value. Figure 15a–d points out that when the inverter transferred low power, the switching loss totally dominated the power loss, and the all-SiC CSI had a great advantage in power loss reduction. This power loss reduction was more pronounced at a high switching frequency. Figure 16a–d showed the curves of the efficiency versus the output power and the switching frequency in the all-Si- and the all-SiC-based CSIs. Figure 16a gives the efficiency curves when the input DC voltage was 100 V and the output power was 100, 150, 200, 250, 300, 350, 400, and 500 W; Figure 16b gives the efficiency curves when the input DC voltage was 400 V and the output power was 2, 2.5, 3, 3.5, 4, and 4.5 kW. Figure 16c shows the efficiency curves when the input DC voltage was 100 V and the switching frequency was 10, 20, 30, 40, and 50 kHz, and Figure 16d shows the curves when the input DC voltage was 400 V. As shown in Figure 16a, under all kinds of load conditions, the efficiencies of the all-SiC-based CSI were all higher than those of the all-Si-based CSI. Figure 16b shows the efficiencies of the all-SiC-based CSI were all higher than those of the all-Si-based CSI.

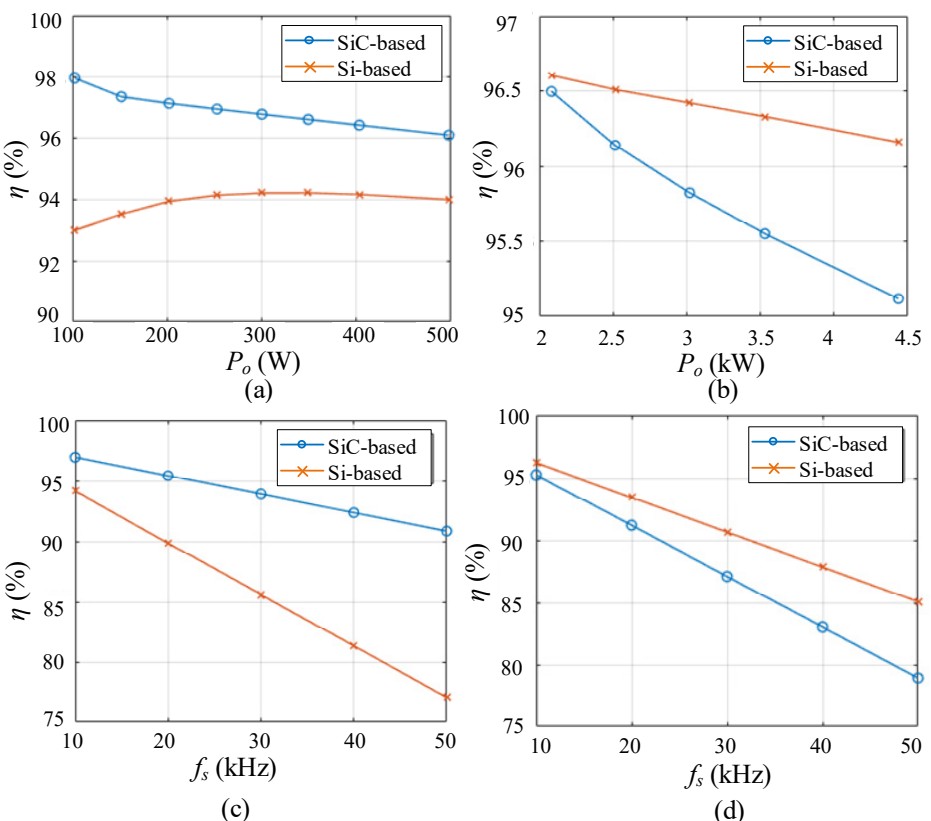

**Figure 16.** The power efficiency comparison between the SiC- and Si-based CSIs: (**a**) the power efficiency versus the output power (when $f_s$ = 10 kHz, $U_{dc}$ = 100 V); (**b**) the power efficiency versus the switching frequency (when $R_{load}$ = 20 Ω, $U_{dc}$ = 100 V); (**c**) the power efficiency versus the output power (when $f_s$ = 10 kHz, $U_{dc}$ = 400V); (**d**) the power efficiency versus the switching frequency (when $R_{load}$ = 20 Ω, $U_{dc}$ = 400 V).

## 4. Simulation and Experimental Verification

To verify the analysis and proposed models about output characteristics and power loss properties, the simulation and experiments were conducted based on the CSI designed in Section 2. Moreover, the parameters in this simulation and experiment are given in Table 5.

**Table 5.** Related parameters in the simulation and experiment.

| Parameter | Value |
|---|---|
| $L$ (mH) | 1.25 |
| $V_{dc}$ (V) | 100, 200 |
| $R_{load}$ (Ω) | 10, 20, 50, 100 |
| $f_s$ (kHz) | 10 |
| $D$ | 0.1~0.9 |
| $f_{sw}$ (kHz) | 100 |

### 4.1. Simulation Results

The simulations were based on PLECS. Actual physical models downloaded in the official website were used in this simulation. The switching frequency of the all-SiC-based CSI was maintained at 10 kHz. The simulation waveforms are shown in Figure 17 where (a) and (b) show the output waveforms when $U_{dc}$ = 100 V and $D$ = 0.5, respectively; (c) and (d) show the waveforms when $U_{dc}$ = 200 V; and (e) and (f) show the results when $D$ = 0.9 and $D$ = 0.1, respectively. Furthermore, Figure 17f,g present the waveforms when the

inverter worked with unbalanced bipolar pulses to verify that the system can work with the reflex charging mode.

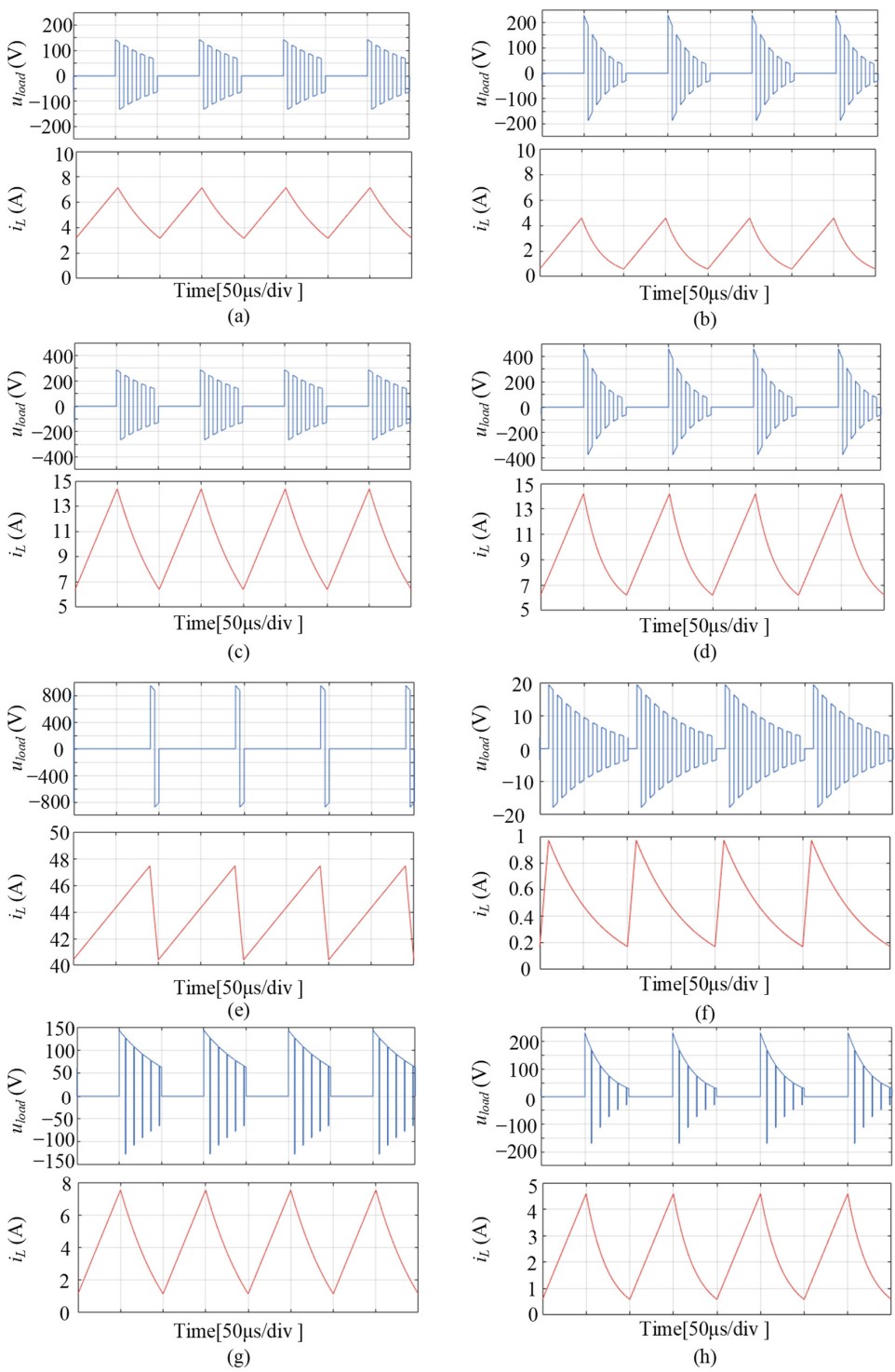

**Figure 17.** Simulation results: (**a**) $U_{dc}$ = 100 V, $R_{load}$ = 20 Ω, $D$ = 0.5; (**b**) $U_{dc}$ = 100 V, $R_{load}$ = 50 Ω, $D$ = 0.5; (**c**) $U_{dc}$ = 200 V, $R_{load}$ = 20 Ω, $D$ = 0.5; (**d**) $U_{dc}$ = 200 V, $R_{load}$ = 50 Ω, $D$ = 0.5; (**e**) $U_{dc}$ = 100V, $R_{load}$ = 20 Ω, $D$ = 0.9; (**f**) $U_{dc}$ = 100 V, $R_{load}$ = 20 Ω, $D$ = 0.1. The reflex charging mode simulations: (**g**) $U_{dc}$ = 100 V, $R_{load}$ = 20 Ω, $D$ = 0.5 and (**h**) $U_{dc}$ = 100 V, $R_{load}$ = 50 Ω, $D$ = 0.5.

### 4.2. Experiments Results

To validate the circuit analysis in Section 2 and the calculation and simulation results in Section 3, a 100 V all-SiC-based CSI and power efficiency testing platform were built in the lab based on the design. The physical picture of the CSI prototype and the test platform are shown in Figure 18. $S_1$–$S_5$ in Figure 18a were C3M0025065D, and the input voltage was supplied by an AMP SP200VDC4000W. The load was non-inductance, the inductance of which was measured to be smaller than 1 μF. The waveforms were collected by a Tektronix MSO46, and the power losses were measured by an HIOKI PW6001.

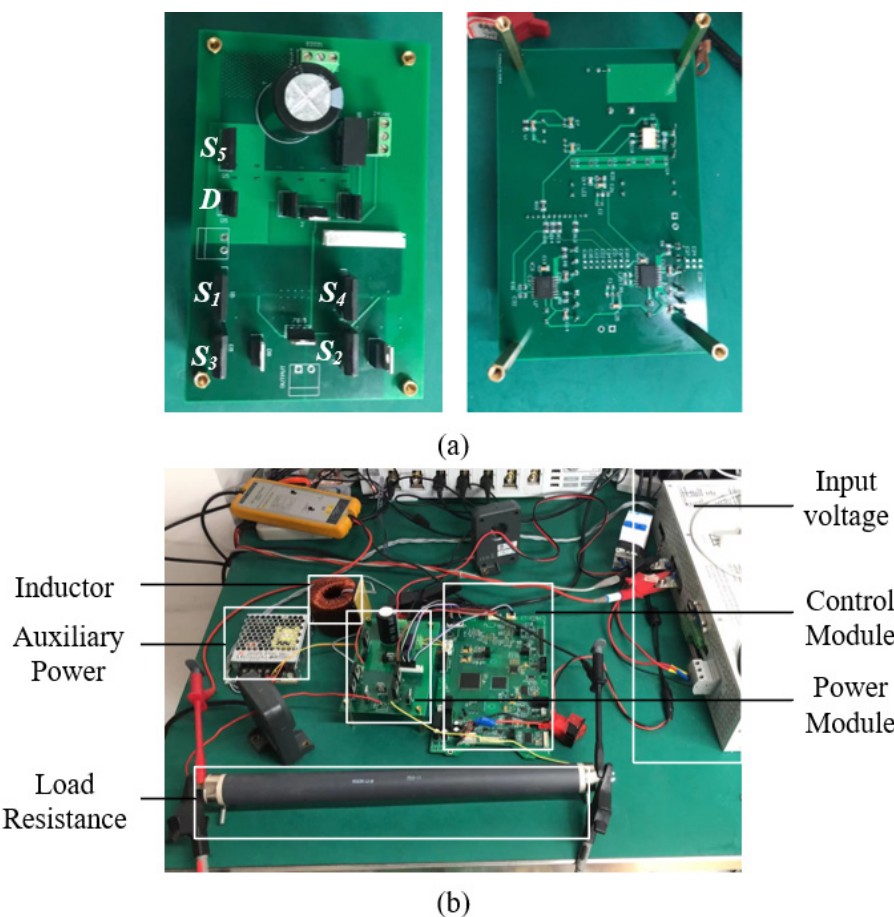

**Figure 18.** Photograph of the CSI and the testing platform: (**a**) power module of CSI; (**b**) efficiency testing platform.

Figure 19 shows the experimental waveforms of the all-SiC-based CSI when the load resistance was 20 and 50 Ω. As it shows, the experimental waveforms agreed with the simulation results, and they were also consistent with the analysis and formulas previously proposed.

The power loss and efficiencies were measured by a power analyzer, and the efficiencies of the CSI were measured when the load resistance was 10, 20, 50, and 100 Ω. Moreover, the inductor losses that were measured were involved. According to the experimental results, the efficiency versus the output power could be plotted as shown in Figure 20. The curves based on the calculated and simulation results are also given for comparison, where the inductor loss of the calculation and simulation results were thought to be equal to the experimental results.

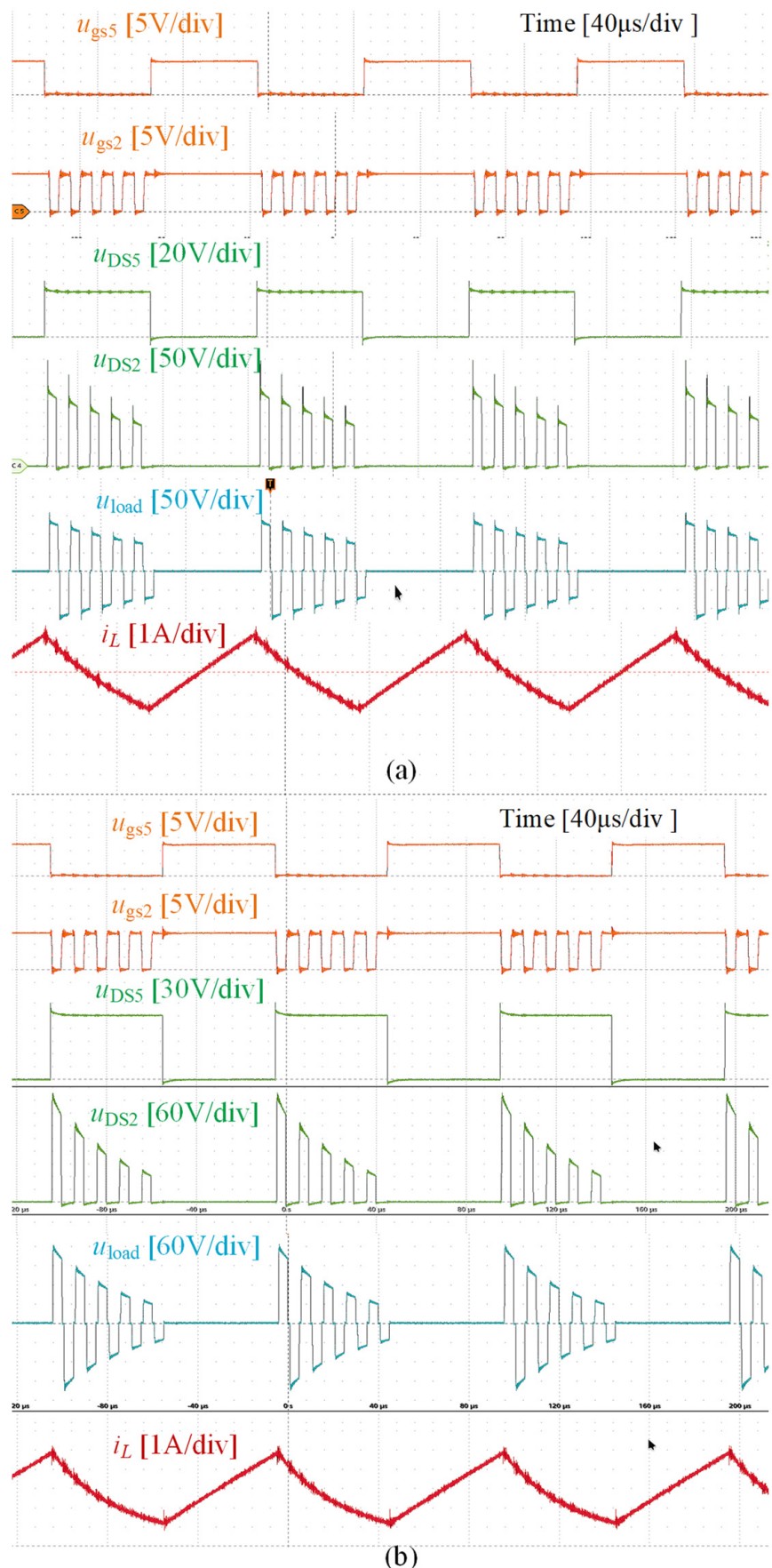

**Figure 19.** The experimental waveforms of the all-SiC-based CSI: (**a**) $R_{load}$ = 20 Ω, and (**b**) $R_{load}$ = 50 Ω.

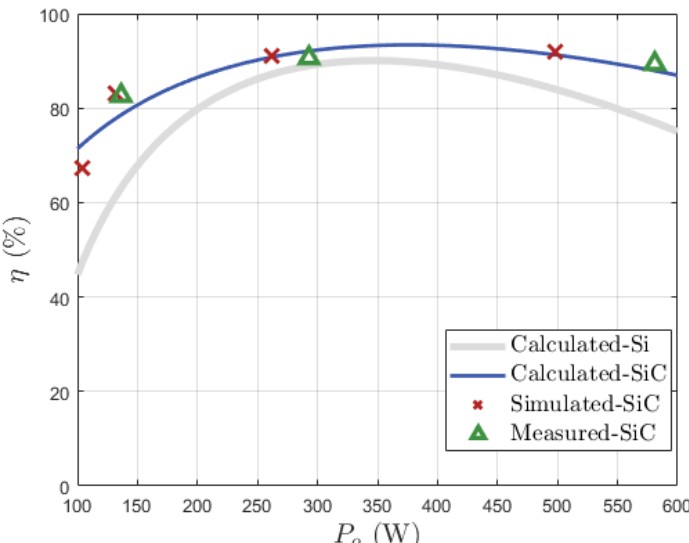

**Figure 20.** The experimental efficiency of the SiC-based CSI, the efficiency simulated by PLECS, and calculated results of the SiC- and Si-based CSIs.

## 5. Conclusions

CSIs have a high MTBF due to their capacitorless features and short-circuit tolerance ability. Therefore, CSIs can be a promising technology for EV chargers. However, the waveforms, the output voltage ripple, and the current ripple have not yet been fully analyzed. This study derived the closed-loop equations of the waveforms to comprehensively understand the operating principles and all the waveforms. Furthermore, based on the derived output ripple equation, this paper points out that the output voltage would encounter the voltage overshoot under a light load condition. Based on the derived equations and the over-voltage condition, this article proposes the parameter design procedure. Furthermore, the power efficiency of both the SiC- and Si-based CSIs were derived. The results showed that SiC-based CSIs could achieve an 11.02% peak efficiency increase, 16.52% light load efficiency increase, and 4.02% full load efficiency increase. Therefore, with the wide use and the fast development of SiC MOSFETs, CSIs will become increasingly attractive in EV chargers. Our simulations and experiments validated the correctness of both the equations and the power conversion efficiency.

**Author Contributions:** Conceptualization, X.Y.; methodology, X.Y. and Z.Z.; software, J.X.; validation, K.L. and C.W.; formal analysis, X.Y.; investigation, K.L.; resources, C.W.; data curation, Z.Z.; writing—original draft preparation, X.Y.; writing—review and editing, K.L. and C.W.; visualization, X.Y.; supervision, K.L. and J.Q.; project administration, K.L.; funding acquisition, K.L. All authors have read and agreed to the published version of the manuscript.

**Funding:** This research was funded by Pioneering Project of Academy for Engineering and Technology of Fudan University (No. gyy2018-002); National Natural Science Foundation of China (No. 51877406).

**Data Availability Statement:** Not applicable.

**Acknowledgments:** The authors would like to thank professors and companions in the Research Team of High Power Electronics, Fudan University, for their experimental assistance.

**Conflicts of Interest:** The authors declare no conflict of interest.

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
