# Peer review of "Analysis and Parameter Design of SiC-Based Current Source Inverter (CSI)"

_wevj, doi:10.3390/wevj13100187_

Round 1

Reviewer 1 Report

The authors of the paper proposed an analysis and parametric design of a SiC-based current source inverter. This method is very clearly presented and documented. Circuit topologies and operating principles and parametric design procedures are well documented. It has been verified through simulations and experiments. My only curiosity is that the author described a current source inverter as an EV application, but used a 100V input in their experimental specifications.

Author Response

Thank you very much for your constructive comments. Please check our point-to-point response in the attachment. 

Reviewer 2 Report

This paper has a fundamental analysis on the CSIs topology with SiC MOSFETs and then discuss the possibility to apply this technique in EV chargers. CSIs are mostly used in medical grade devices because of the high reliability and the capacitor-less feature, but for EV chargers, operation-modes, waveforms, parameter designs, etc., are vague. This paper not only addresses all the issue with analytical solutions, but also points out that CSIs can encounter over-voltage phenomenon in EV chargers during light load. The simulations and experiments are sufficient to verify all the details of the derived equations.

However, some minor issues should be corrected:

1. In Figure 11, the meaning of all the dots has never been explained. It is important why some dots exist.

2. The term, “all Si” should “all-Si”in Figure 7.

3. Figure. 7’s quality should be improved. The Y-axis is confusing. Why iD(t) is the origin in the first figure in Figure. 7?

4. The authors should include the details of the measurement instruments. It would improve the reproducibility of experiments.

Author Response

(The authors gave the same response as above.)
